# The impact of environmental regulatory instruments on agribusiness technology innovation—A study of configuration effects based on fsQCA

Jinglin Xia⊙*[☉], Liguo Zhang[☉], Yuwei Song⊙[☉]

School of Economics, Jiangxi University of Finance and Economics, Nanchang, Jiangxi, People's Republic of China

☉ These authors contributed equally to this work.
* xiajinglin@jxufe.edu.cn

## Abstract

This paper investigates the complex causal relationships between various types of environmental regulatory instruments (ERI) and agri-firms' technological innovation employing fuzzy set qualitative comparative analysis (fsQCA). The study finds a well-designed set of ERI can promote technological innovation in agribusiness; control-command ERI cannot promote technological innovation in agribusiness solely, market-incentivized ERI is indispensable in promoting firms' innovation performance, implicit ERI plays an important role in promoting firms' innovation and voluntary ERI does not play a significant role in promoting firms' technological innovation. The government should coordinate among various types of ERI and improve the design of ERI to achieve a win-win situation for both economic and environmental performance in the agriculture sector.

## Introduction

Global agricultural production has experienced unprecedented growth in recent decades. Countries are striving to improve their agricultural science and technology, promote the modernization and transformation of agriculture and rural areas, and enhance the core competitiveness of national agricultural industries. However, traditional agricultural production has generated negative externalities for the natural environment due to the application of pesticides and fertilizers, the emission of livestock manure, and the excessive consumption of resources. Agriculture and climate change are intertwined and have an impact on one another [1]. Promoting environmentally friendly agricultural technology innovation is a significant way to realize the transformation of rural growth patterns, sustainable rural development and agricultural economic growth. In recent years, the government and society have placed a great importance on the subject of green development in agriculture [2]. Countries have established various environmental protection agencies and introduced a series of environmental regulation instruments (hereinafter referred to as "ERI"), such as environmental protection acts, environmental protection taxes and fees and environmental protection initiative conventions.

**Funding:** We acknowledge the financial support from: National Natural Science Foundation of China (Grant No. 72073054) for L. Zhang; Jiangxi Provincial Association of Social Sciences (Grant No. 22WT60) for J. Xia, & L. Zhang; Provincial Social Science Foundation of Jiangxi (Grant No. 22ZXQH07) for L. Zhang; 2022 Postgraduate Innovation Special Foundation of Jiangxi (Grant No. YC2022-B135) for J. Xia. The funders had no role in study design, data collection and analysis, decision to publish, or preparation of the manuscript.

**Competing interests:** The authors have declared that no competing interests exist.

Those have performed an effective role in combating environmental pollution, but how they affect agribusiness technological innovation is still under heated discussion. The analysis of the impact of ERI on technological innovation in agribusiness, the internalization of environmental costs through reasonable environmental regulations, and the realization of a win-win situation for both environmental protection and economic development are important foundations for the modernization and transformation of agriculture and the rational formulation of environmental management policies in countries around the world and are necessary for the construction of a better living environment for human beings. From the results applicable, there is a wealth of research on the role of ERI and the factors influencing enterprise innovation, but there is a lack of research focusing on agricultural enterprises and carried out from the perspective of the configurations of ERI.

The question of whether environmental regulation can promote technological innovation in firms has been relatively well discussed in existing research, but there is no unified view. Meanwhile, there aren't as many research focused on the connection between environmental regulation and technological innovation in the context of agribusiness. Prior studies have argued environmental regulation has a disincentive effect on firms' technological innovation. Such studies focused on the costs of environmental regulation, arguing that environmental regulation can lead to higher production costs, reduced competitiveness and disincentives to innovate by imposing more stringent emission standards, additional taxes and administrative costs, and crowding out investment in innovative projects [3–5]. The Porter hypothesis, represented by the concept of "innovation compensation", argues that well-designed environmental regulations can promote technological innovation [6]. The case studies of the Toyota Prius and Nissan Leaf amply supported the idea that environmental rules spur innovation among Japanese businesses [7]. The hypothesis has also been tested in empirical studies in developed economies [8, 9]. In emerging economies, a stricter environmental regulatory framework not only combats pollution but also shifts the innovation activities of manufacturing firms towards building a knowledge base on environmental protection [10]. In the case of agricultural firms, it has been elucidated that innovative technologies have improved environmental protection [11]. Some scholars tested the "strong Porter hypothesis" and "weak Porter hypothesis" by measuring green total factor productivity in agriculture under environmental regulation, which holds for agricultural production in China [12]. Under environmental regulations, both total factor productivity and environmental technical efficiency in agriculture have increased and improved during the transition period [13]. It has also been argued that in the short term, environmental regulation is detrimental to technological progress in agriculture. But in the long term, environmental regulation is beneficial to agricultural technological progress [14]. In addition, some studies have argued the effect of environmental regulation is related to the intensity of the regulation. There is also a class of views that the impact of environmental regulation on technological innovation is non-linear [15, 16]. In an empirical study of pollution-intensive industries, it was found that current-period environmental regulations negatively affect environmental efficiency, but lagged-period environmental regulations promote environmental efficiency, suggesting the "compliance cost" of current-period environmental regulations to pollution-intensive firms is greater than the "innovation compensation" effect [17]. The possible reason for the inconsistency of the many views above lies in the fact that early environmental regulation was mainly implemented by the government, and the ignorance of changes in the behavior of businesses and the public as the leading actors. Some studies also use simple indicators to represent environmental regulation for comparative analysis between regions, reflecting only the overall effect of various ERI when they work together and reducing sensitivity to differences in specific regulatory instruments.

With the increasing emphasis on environmental governance and the gradual development of environmental regulation, the classification of ERI has gradually evolved from a dichotomous approach [18] to a trichotomous approach [19] and a quadratic approach [20]. Among these types of environmental regulation, the subjects of control-command and market-incentivized ERI represent the government, while the direct subjects of voluntary and implicit ERI are not the government, but the regulatory effects of both depend on the guidance of government policies and regulations, technical standards and other supervision and management systems, providing a theoretical basis for this paper's idea of incorporating the four types of ERI into the same research framework [21]. Regarding the role of various ERI, current research has primarily focused on the effects of individual or single types of regulatory instruments, or the comparison of the effects between several regulatory instruments. Regarding individual ERI, some scholars argue the "innovation compensation" effect of market-based incentives is not significant, and that environmental investment has a significant positive effect on output growth [22]. It has been verified that specific environmental regulatory policies significantly contribute to R&D inputs and innovation outputs in agro-biofuel technologies [23]. It has also been discussed that environmental taxes promote spillovers from heavy pollution in China, while control-command ERI is more targeted at technological innovation in energy efficiency and emission reduction [24]. However, the effect of trade and environmental policies on each developing country's green innovation is likely to vary according to the maturity of the political institutions [25]. Although emissions trading mechanisms cannot play a role in reducing emissions in the short's, they can drive significant emission reductions in the long term [26].

In summary, the efficiency of ERI and their ability to promote technological innovation in agricultural firms, especially the differences in the impact of various ERI on firms' technological innovation and the effects of the implementation of different regulatory instrument configurations, remain controversial, while the latter has not been fully discussed. The previous studies were primarily conducted through traditional linear regression models, while environmental regulations are interactional rather than independent. Therefore, it was not possible to derive a complex conclusion from the causal relation using only quantitative approaches. Recently, scholars have shifted to explore the paths for a multi-policy combination to influence the outcome by applying QCA. It serves as a bridge between quantitative and qualitative methods, ideally balancing the depth of case study knowledge supplied by qualitative analysis with the breadth of analysis provided by quantitative data [27]. Based on this, this paper selects 52 agricultural A-share listed firms for the study and uses fuzzy set qualitative comparative analysis (fsQCA) to explore the complex causal relationships between various environmental regulatory instruments and agri-firms' technological innovations, as well as the fit between various ERI based on a configuration perspective. The possible marginal contributions of this paper are (1) the application of fuzzy set qualitative comparative analysis (fsQCA) in this study. Although scholars have conducted numerous studies on the heterogeneity of ERI and the impact of multiple instruments on firms' technological innovation, they have generally treated each regulatory instrument as an independent factor and neglected the impact of various types of ERI acting together on agricultural technological innovation. QCA has a hybrid nature that combines the benefits of case-based (qualitative) and variable-oriented (quantitative) methodologies [28]. The application of fsQCA can fill the gap of existing studies; (2) the focus on agricultural industries. The current research mostly focuses on industry, high-tech enterprises or heavy pollution industries. The focus on agricultural technology innovation is relatively late compared to other industries, and there are very few studies on the impact of ERI on technological innovation in agricultural enterprises. This paper selects agricultural A-share listed enterprises as a research sample to provide a theoretical reference for how environmental regulatory instruments affect technological innovation in agricultural enterprises.

# Research methodology and content

## Research methodology

Qualitative comparative analysis (QCA) is a Boolean algebra-based approach to set-theoretic configuration state analysis that enables a holistic exploration of 'how' complex social problems induced by multiple concurrent causalities occur by examining sufficient and necessary subset relationships between antecedent conditions and outcomes [29]. As a methodological innovation, QCA conceptualizes causality as complex causation characterized by conjunction, equivalence and asymmetry [30]. It integrates the strengths of case studies and variable studies and helps to resolve complex causal problems such as multiple concurrent causalities, causal asymmetry, and multiple scenario equivalence [31]. The method empirically investigates each and all combinations (high/low or absent) of the outcome of interest (in the present case, Agri-firms' innovation performance) and its predictors (various environmental regulation instruments).

Some of the endogeneity problems prevalent in traditional symmetric quantitative methods do not exist in the QCA method [32]. The endogeneity problems of traditional regression methods mainly stem from the assumption of equilibrium and normal distribution, and the focus on linear, symmetric relationships of independent variables. The QCA method avoids reverse causality at source based on a configuration perspective and causal asymmetry. And because it is based on Boolean algebra and examines aggregated rather than correlated relationships, the QCA approach is also not subject to omitted variable bias. Finally, QCA's retro-causal inference and ensemble analysis do not rely on random sampling and therefore do not suffer from the sample selection bias associated with random sampling.

Depending on the type of variables, QCA is divided into clear set qualitative comparative analysis (csQCA), multi-valued set QCA (mvQCA) and fuzzy set QCA (fsQCA). Of these, mvQCA and csQCA share a common methodological basis of clear sets and truth-tables, which dictate they only suitable for categorical problems. fsQCA enhances the ability to analyze fixed-distance, fixed-ratio variables and allows QCA to also deal with problems of varying degrees and problems of partial affiliation, where cases have an affiliation score between 0 and 1. And by converting the fuzzy set data into a truth table, fsQCA retains the advantages of truth table analysis in dealing with qualitative data, limited diversity and simplified configuration, giving fsQCA the dual properties of qualitative and quantitative analysis [33]. Given the properties of the methods and the characteristics of the data selected, this paper applies fsQCA for analysis.

## Data sources

As the data disclosed by listed enterprises are more comprehensive and have better credibility, and the classification of stock market industry sectors is relatively clear, this paper selected 52 listed enterprises from the Sector of Agriculture, Forestry, Animal Husbandry and Fishery in Shanghai and Shenzhen A-shares, excluding ST enterprises, and combined with the completeness of enterprise annual reports, as the research sample. The data used in the paper were obtained from the annual reports of listed enterprises, the Guotai'an (CSMAR) database, the patent search platform of the National Intellectual Property Office, the National Certification and Accreditation Information Public Service Platform, the China Environment Statistical Yearbook, the China Statistical Yearbook, all for the year 2020.

## Condition selection

**Outcome condition.** The amount of granted invention patents per R&D personnel (PTT). In existing studies, the use of patent grants represents a more conventional method

to portray the innovation output of enterprises [24, 34]. Referring to the indicators used by the Ministry of Science and Technology and some localities for the evaluation of innovative enterprises, this paper selects the number of authorized invention patents owned by R&D personnel per capita as the outcome condition. The number of authorized patents for inventions owned by the enterprise as the patentee, the number of invention patents authorized by domestic and foreign patent administration departments and within the validity period were used. The number of R&D personnel adopts the data published in the 2020 annual report.

**Causal conditions.**

1. Choose the environmental governance penalty (PNT) and the state ownership of the enterprise (NTL) in the area to denote the Control-command type ERI. The higher the number of environmental governance penalties per capita, the higher the governance effort. For the control-command instrument, state-owned enterprises are an important policy transmission channel. The effect of control-command ERI is stronger in industries with a higher degree of nationalization [24, 35]. It has also been suggested that SOEs tend to apply for high-quality green invention patents, while firms in highly polluting industries tend to apply for low quality green utility model patents [36]. Therefore, this paper collates the ratio of the number of annual environmental governance penalty cases to the total population of the region in which the enterprise is located, as well as the criteria for defining state-owned and state-controlled enterprises and enterprises under the effective control of the state according to Article 4 of the Measures for the Supervision and Administration of State-owned Assets Transactions of Enterprises, based on the shareholders disclosed in the enterprise's annual report, to determine whether it is a state-owned or state-controlled enterprise.

2. Choose environmental taxes (TAX) (in thousand yuan) and environmental subsidies (SSD) (in thousand yuan) to represent market-incentivized ERI. Taxes and subsidies are important ways for the government to act as a "visible hand" in market-incentivized environmental regulation. This paper uses the ratio of environmental protection tax revenue to regional GDP in the region where the company is based, as well as the number of government-related environmental subsidies manually collated from the company's annual report, to characterize the strength of market-incentivized ERI.

3. Whether the selected enterprises are certified to ISO 14001 environmental management system (ISO) is selected to represent voluntary type ERI [21]. This paper measures the strength of voluntary environmental regulation based on whether the sample enterprises are certified to ISO 14001 environmental management system and quantifies it using a dichotomous assignment method, i.e., enterprises are assigned a value of 1 if they are certified, and 0 if they are not.

4. Choose online exposure (EXP) (in thousand) to reflect the intensity of implicit ERI. Media monitoring, netizen supervision and public opinions pressure are all forms of implicit environmental regulation [37]. In this paper, we employ the number of search results related to the enterprise from the head searching engine to represent the amount of social attention and the intensity of implicit ERI faced by the enterprise on the Internet.

The descriptive statistics for each conditions selected are shown in Table 1.

## Data calibration

The fsQCA method requires calibration to convert the raw data into fuzzy affiliation values between 0 and 1 for the further step of the analysis. For fuzzy set calibration, existing research

**Table 1. Descriptive statistics.**

| | Outcome conditions | Causal conditions | | | | | |
| | | Command-control | | Market-incentivized instrument | | Voluntary | Implicit |
| | **PTT** | **PNT** | **NTL** | **TAX** | **SSD** | **ISO** | **EXP** |
| Average | 0.446 | 0.305 | 0.423 | 2.051 | 3107.694 | 0.192 | 8981.346 |
| Median | 0.182 | 0.147 | 0.000 | 1.670 | 646.295 | 0.000 | 4285.000 |
| Std Dev. | 0.974 | 0.363 | 0.499 | 1.307 | 6156.411 | 0.398 | 14997.456 |
| Minimum | 0.000 | 0.024 | 0.000 | 0.637 | 0.000 | 0.000 | 1260.000 |
| Maximum | 6.449 | 1.338 | 1.000 | 7.199 | 26598.506 | 1.000 | 100000.000 |
| Counts | 52 | 52 | 52 | 52 | 52 | 52 | 52 |

has used two main calibration methods—the indirect calibration method requires the researcher to assign a value between 0 and 1 for each condition based on personal judgement [33]; the direct calibration method proposes three qualitative anchor points based on theory and practice, i.e., fully affiliated, fully unaffiliated and intersection points, then calibrated using an algorithm provided by the software [38]. The direct calibration method adopts statistical models, is more formal and is the most commonly used calibration method [39]. Therefore, the direct calibration method is applied in this paper. The calibration anchor points for each variable selected in this paper are shown in Table 2. 95%, 50% and 5% quartiles are commonly used as calibration anchor points for normally distributed data in general, but since the data in this paper generally show right skewness and the mean is greater than the median, PTT, PNT, TAX, SSD, and EXP all use 85% quartiles as fully affiliated points, medians as crossover points, and 15% quartiles as fully unaffiliated points. NTL and ISO are dichotomous assignments and do not require more isolated calibration. The calibration anchors for each condition are shown in Table 2.

## Analysis of the empirical results

### Necessary conditions analysis

Before performing the fsQCA criteria analysis, a necessary condition analysis is performed [38]. A "necessary condition" can be considered as a superset of the results, which must not occur if the condition does not exist and should be excluded from the truth table analysis. A consistency score of 0.9 and adequate coverage is ordinarily required to identify a necessary condition [40]. Necessity was analyzed through the fsQCA 3.1 software and the results are shown in Table 3. The consistency values for the cause conditions alone are all below 0.9, indicating none of the individual factors constitutes a necessary condition and that one ERI alone does not guarantee high innovation performance for the firm.

**Table 2. Calibration anchors.**

| | | Fully affiliated points | Intersection point | Fully unaffiliated points |
|---|---|---|---|---|
| Result condition | PTT | 0.718 | 0.182 | 0.000 |
| Cause conditions | PNT | 0.754 | 0.147 | 0.043 |
| | TAX | 3.370 | 1.670 | 0.880 |
| | SSD | 5082.100 | 646.295 | 0.000 |
| | EXP | 17300.000 | 4285.000 | 2251.000 |

**Table 3. Results of necessity analysis of individual conditions.**

|  | PTT | | ~PTT | |
|---|---|---|---|---|
|  | **Consistency** | **Coverage** | **Consistency** | **Coverage** |
| PNT | 0.530 | 0.563 | 0.453 | 0.568 |
| ~PNT | 0.594 | 0.479 | 0.652 | 0.621 |
| NTL | 0.359 | 0.390 | 0.477 | 0.610 |
| ~NTL | 0.641 | 0.509 | 0.523 | 0.491 |
| TAX | 0.657 | 0.628 | 0.483 | 0.545 |
| ~TAX | 0.524 | 0.462 | 0.670 | 0.697 |
| SSD | 0.610 | 0.615 | 0.475 | 0.564 |
| ~SSD | 0.567 | 0.478 | 0.676 | 0.672 |
| ISO | 0.197 | 0.471 | 0.188 | 0.529 |
| ~ISO | 0.803 | 0.456 | 0.812 | 0.544 |
| EXP | 0.600 | 0.606 | 0.492 | 0.592 |
| ~EXP | 0.597 | 0.501 | 0.670 | 0.664 |

*Note*: "~" means the logical operation "not", that is, the condition does not exist.

## Analysis of condition configurations

To ensure the adequacy of the conditional configurations, the consistency threshold for each configuration was set to the commonly recommended value of 0.8, and the case frequency was set to 1, considering that more than 75% of the observed cases needed to be retained. fsQCA 3.1 software was used to figure out the complex, intermediate and parsimonious solutions. The complex solution is somewhat one-sided as it does not incorporate logical residuals, i.e., it does not consider unobserved cases. The parsimonious solution underwent a simple and complex counterfactual analysis with a minimal number of configurations and conditions. The intermediate solution considers a simple counterfactual analysis, incorporating logical residual terms that are consistent with theoretical direction expectations and empirical evidence. Therefore, the analysis is usually conducted only for parsimonious and intermediate solutions [39]. When a condition appears in both the parsimonious and intermediate solutions, it is a core condition, and when it appears only in the intermediate solution it is a marginal condition. The core condition plays a primary role, and the edge condition plays a secondary role. In addition, to reduce possible contradictory configurations, the PRI consistency was set to 0.75 [41]. The final results are presented in Table 4.

As shown in Table 4, 6 ERI configurations produce high innovation performance (configurations I-VI) and 3 ERI configurations produce non-high innovation performance (configurations VII-IX). The consistency of the solutions for each configuration in the table and the overall consistency are above 0.8, indicating that each configuration is a sufficient condition for the outcome.

**Configuration of ERI that produce high innovation performance.** Configuration I (PNT×NTL×TAX×~SSD×ISO×EXP) consists of the core conditions of high environmental taxation, voluntary environmental quality certified, high public exposure, and the marginal conditions of high environmental penalty intensity, state-controlled and non-high environmental subsidies. A consistency score of more than 0.9 for this configuration is a necessary condition. Agri-listed firms face strong instruments of control-command type environmental regulation, while in the context of strict market-type regulation with high taxes and low subsidies, effective public scrutiny and better environmental initiatives by the firms themselves are

**Table 4. fsQCA analysis of ERI configurations.**

| | PTT | | | | | | ~PTT | | |
|---|---|---|---|---|---|---|---|---|---|
| | I | II | III | IV | V | VI | VII | VIII | IX |
| PNT | • | • | • | ⊗ | ⊗ | ⊗ | ● | ⊗ | • |
| NTL | • | ⊗ | ⊗ | ● | ● | ⊗ | ● | | • |
| TAX | ● | ● | ● | ● | ⊗ | • | | ⊗ | ⊗ |
| SSD | ⊗ | ● | ⊗ | ⊗ | ● | • | ⊗ | ⊗ | • |
| ISO | ● | | ⊗ | ⊗ | ⊗ | ⊗ | ⊗ | ⊗ | ● |
| EXP | ● | ● | ⊗ | ● | ● | | | ⊗ | ⊗ |
| Consistency | 0.974 | 0.871 | 0.882 | 0.889 | 0.993 | 0.845 | 0.977 | 0.899 | 1.000 |
| Raw Coverage | 0.032 | 0.167 | 0.132 | 0.098 | 0.058 | 0.151 | 0.153 | 0.215 | 0.036 |
| Unique coverage | 0.032 | 0.078 | 0.079 | 0.063 | 0.024 | 0.061 | 0.123 | 0.185 | 0.036 |
| Solution consistency | 0.853 | | | | | | 0.933 | | |
| Solution coverage | 0.463 | | | | | | 0.373 | | |

Note: "●" indicates that the core condition exists, "⊗" indicates that the core condition does not exist, "•" indicates that the edge condition exists, "⊗" indicates that the edge condition does not exist, and blank indicates that the condition may or may not exist. Conditions that appear in the parsimonious solution are referred to as core conditions in the configuration, indicating a strong causal relationship with the outcome of interest. The remaining conditions that appear in the intermediate solution but not in the parsimonious solution are called edge conditions and have a weaker causal relationship with the outcome.

required to jointly contribute to the effectiveness of good technological innovation. This may be due to the public nature of agribusiness technology innovation products, which have significantly lower economic returns than non-agricultural enterprises [42], therefore in the presence of high negative economic incentives, stronger monitoring and better environmental awareness by firms are generally required to motivate firms to innovate.

Configuration II (PNT×~NTL×TAX×SSD×EXP) consists of the core conditions of high environmental taxes, high environmental subsidies, and strong implicit regulation, and the marginal conditions of high environmental penalty intensity and non-state ownership. The government provides a strict regulatory environment by enforcing high-intensity environmental enforcement and market-incentivized environmental regulation, while non-state capital control of firms provides a relatively flexible space for technological innovation and a strong environmental awareness of society at large creates implicit regulatory pressure on firms. It can also be inferred that non-state enterprises are more likely to be able to achieve high levels of innovation when all types of ERI are of high intensity.

Configuration III (PNT×~NTL×TAX×~SSD×~ISO×~EXP) consists of the core conditions of non-state holding, high environmental tax, non-voluntary guilty plea, and non-high exposure, and the marginal conditions of high environmental penalty intensity and non-high environmental subsidy. In this scenario, firms achieve high innovation output. Agricultural firms have less core incentive to innovate compared to other non-agricultural firms, and in the absence of state capital support and higher environmental subsidies and public scrutiny, stricter environmental penalty policies and higher environmental taxes exert more pressure on firms to increase technological innovation to reduce costs or improve efficiency.

The core conditions for Configuration IV (~PNT×NTL×TAX×~SSD×~ISO×EXP) include non-high penalty intensity, state-controlled, high environmental taxation, and high exposure, with marginal conditions of non-high environmental subsidies and non-high voluntary certification. State-controlled businesses with significant exposure can nonetheless produce a lot of innovation even in environments where there is little environmental

enforcement pressure, non-high environmental initiatives, and high taxation few economic incentives. The more socially visible state-owned agribusinesses do not have a strict penalty system for environmental violations or favorable policies in terms of environmental taxes and subsidies, but their social responsibilities make them better examples of how to foster technological innovation to reduce costs, improve operational efficiency and maintain their massive operational needs.

Configuration V (~PNT×<u>NTL</u>×~TAX×<u>SSD</u>×~ISO×<u>EXP</u>) consists of a combination of the core conditions of state-controlled, high environmental subsidies, and high social monitoring, and the marginal conditions of non-high environmental penalties, non-high environmental taxes, and non-high voluntary-type regulation. Similar to configuration IV, state-controlled agricultural firms with high exposure can achieve high technological innovation output in the absence of high-intensity control-command and voluntary-type environmental regulation, regardless of whether they face positive or negative market-incentivized ERI.

Configuration VI (~PNT×~NTL×TAX×SSD×~ISO) consists of the marginal conditions of non-high environmental penalty, non-state holding, high environmental taxation, high environmental subsidy, and non-high voluntary environmental certification. It indicates that when neither control-command nor voluntary ERI are strong, high-intensity market-incentivized ERI can still lead to high innovation output by firms. This configuration suggests that firms can achieve better technological innovation outcomes when all types of ERI are weak and only market-incentivized ERI are strong.

**Configuration of ERI that produce non-high innovation output.** Configuration VII (<u>PNT</u>×<u>NTL</u>×~<u>SSD</u>×~<u>ISO</u>) is a necessary condition with a consistency score greater than 0.9 and consists of the core condition of high environmental penalties, state-controlled, non-high voluntary type regulation and the marginal condition of non-high environmental subsidies. It shows that with weak market-incentivized-based, voluntary, and implicit environmental regulation, the mere presence of control-command type regulatory instruments cannot support agribusiness to achieve high innovation output, regardless of high environmental taxes and high implicit environmental regulation.

Configuration VIII (<u>~PNT</u>×<u>~TAX</u>×<u>~SSD</u>×~ISO×~EXP) contains the core conditions non-high environmental penalties, non-high environmental taxes, non-high environmental subsidies, non-high implicit regulation, and the marginal condition non-high voluntary type regulation. When all five ERI, including environmental governance penalties, environmental taxes, environmental subsidies, voluntary certification, and internet exposure, are weak, neither state-owned nor non-state-controlled agricultural enterprises can achieve high technological innovation effectiveness.

Configuration IX (PNT×NTL×~<u>TAX</u>×SSD×<u>ISO</u>×~<u>EXP</u>) consists of the core condition of non-high environmental taxes, high voluntary-type regulation, and strong implicit regulation, and the marginal condition of high environmental penalties, state-owned control, and high environmental subsidies, with a consistency score of 1.000, which is a necessary condition for firms to have non-high innovation output. This suggests that state-owned agricultural firms with proactive environmental awareness are unable to achieve high innovation output in the absence of public scrutiny and in the face of stringent environmental penalties imposed by local governments, even with positive market-incentivized from high environmental subsidies and non-high environmental taxes.

## Comprehensive analysis between configurations

The potential relationship between conditions can be better understood by the cross-sectional comparison between each configuration solution (Ma & Zhang, 2022).

**Cross-sectional comparison of individual conditions.** High environmental taxation (TAX) was present as a core condition in 4 high innovation output configurations and absent as a core condition in 2 non-high innovation output configurations. High social exposure (EXP) is present as a core condition in the four high innovation output configurations and absent as a core or marginal condition in the two non-high innovation output configurations. This suggests that although these two conditions do not enhance firms' innovation effectiveness on their own, the combination of high environmental taxes and strong public scrutiny are important links in the combined conditions of corporate environmental regulation, and both are closely related to high innovation output. Voluntary environmental quality certification (ISO) is generally absent from the high innovation output configuration. This suggests that voluntary environmental certification does not significantly increase firms' innovation output. Strong environmental penalties, high environmental subsidies, and state ownership as a separate cause condition all score poorly in terms of consistency, and no significant patterns emerge in the other high innovation output configurations. Therefore, high environmental subsidies should not be pursued as a single cause in promoting firms' technological innovation, and there is no significant causal relationship between the two.

**Comparison of configurations.** Compare Configurations I and VII. When both high environmental taxes and high social concern are present, the sole difference between Configuration VII and Configuration I is the presence or absence of voluntary environmental regulation. In this case, a high level of voluntary environmental regulation is associated with a high level of innovation output, but not with a high level of innovation output otherwise. Thus, when all types of environmental regulation instruments are, generally strong but environmental subsidies are not high, the environmental initiative of the firm can determine the outcome of its innovation output.

Compare Configurations IV and V. There is a potential substitution relationship between high environmental taxes and high environmental subsidies for less environmentally proactive state-owned enterprises in a more relaxed environmental enforcement environment and high social concern. The presence and only the presence of one of the two leads to high innovation output by firms. This may be because taxes and subsidies are incentives in opposite directions, and both strong positive incentives (i.e., low taxes and high subsidies) and negative incentives (high taxes and low subsidies) can contribute to higher innovation output by agricultural firms in certain circumstances. However, the presence or absence of both high taxes and high subsidies counteract each other's incentive effects.

Compare Configurations VI, VII and VIII. When none of the conditions that may or may not be present in 3 three Configurations is present, it can be interpreted as having only strong market-incentivized, only strong control-command regulation, or none of the regulatory instruments being strong. In this case, only Configuration VI can lead to high innovation output, i.e., the firm cannot achieve high innovation output when all types of environmental regulation instruments are weak, or when only strong control-command regulation instruments are available. High innovation output can be achieved when only strong market-incentivized are at play.

Compare Configurations I to VI. High environmental taxes and high environmental subsidies are present in either one or the other of all the configurations of causal conditions that lead to high innovation output outcomes. It can be deduced that high environmental taxes and high environmental subsidies cannot be absent at the same time to promote high technological innovation outcomes for firms, i.e., market-incentivized ERI are essential.

## Robustness test

Checking the robustness of the analysis results is a key step in fsQCA studies. fsQCA robustness checks are commonly conducted by reasonably adjusting the settings of relevant

parameters, like calibration basis, minimum case frequency and consistency threshold, and then analyzing the adjusted data again to compare changes in the configurations to assess the reliability of the results [43]. If the adjustment of the parameters did not result in substantial changes in the number of histories, their components, as well as in consistency and coverage, the results of the analysis can be considered reliable [41].

Given the small sample size of the data in this paper, the minimum case frequency needed to be retained was set to 1 to ensure that retained cases were still more than 75% of the original data at the time of testing. Therefore, by adjusting the data calibration anchors to the 80% quantile, median, and 20% quantile as well as the 90% quantile, median, and 10% quantile, the consistency thresholds were adjusted to 0.75 and 0.85 and the PRI consistency thresholds were adjusted to 0.8 and 0.85, and the data were again measured and analyzed. The results of the tests are presented in Table 5.

As seen in Table 5, after adjusting the data calibration anchor points, consistency threshold thresholds and PRI consistency thresholds, the solution terms and configurations derived from the operations did not change or were logical subsets or superset relations of the initial

**Table 5. Robustness test.**

| | | Calibration Anchor Point/ Consistency Threshold/PRI Threshold | Number of configurations | Solution Consistency | Solution Coverage | Configuration differences |
|---|---|---|---|---|---|---|
| (PTT) High innovation | Baseline | 0.15/ 0.8/0.75 | 6 | 0.853 | 0.463 | |
| | Changing calibration anchor | 0.1/0.8/ 0.75 | 4 | 0.867 | 0.267 | Increase in consistency, decrease in coverage, a logical subset of the baseline solution |
| | | 0.2/0.8/ 0.75 | 6 | 0.847 | 0.448 | Decrease in consistency and coverage |
| | Changing consistency threshold | 0.15/ 0.75/ 0.75 | 6 | 0.853 | 0.463 | None |
| | | 0.15/ 0.85/ 0.75 | 6 | 0.855 | 0.416 | Increase in consistency, decrease in coverage, a logical superset of the baseline solution |
| | Changing PRI threshold | 0.15/ 0.8/0.7 | 6 | 0.855 | 0.483 | Increase in consistency and coverage, a logical superset of the baseline solution |
| | | 0.15/ 0.8/0.8 | 5 | 0.903 | 0.313 | Increase in consistency, decrease in coverage, a logical subset of the baseline solution |
| (~PTT) Non-high innovation | Baseline | 0.15/ 0.8/0.75 | 3 | 0.933 | 0.373 | |
| | Changing calibration anchor | 0.1/0.8/ 0.75 | 4 | 0.919 | 0.43 | Decrease in consistency, increase in coverage, a logical subset of the baseline solution |
| | | 0.2/0.8/ 0.75 | 3 | 0.923 | 0.356 | Decrease in consistency and coverage |
| | Changing consistency threshold | 0.15/ 0.75/ 0.75 | 3 | 0.933 | 0.373 | None |
| | | 0.15/ 0.85/ 0.75 | 4 | 0.923 | 0.416 | Decrease in consistency, increase in coverage, a logical superset of the baseline solution |
| | Changing PRI threshold | 0.15/ 0.8/0.7 | 4 | 0.923 | 0.416 | Decrease in consistency, increase in coverage, one more configuration emerged (~PNT×~NTL× ~TAX×~SSD×~ISO) |
| | | 0.15/ 0.8/0.8 | 3 | 0.933 | 0.373 | None |

results, and the consistency and coverage increased or decreased slightly, which did not constitute a fundamental change to the original analysis results. Therefore, the conditional configuration analysis in this paper passes the robustness test.

## Insights and recommendations

### Conclusions

1. This paper confirms the Porter hypothesis that well-designed environmental regulation instruments can facilitate technological innovation. High innovation output in listed agricultural firms cannot be achieved by any one of the six ERI discussed in this paper alone; rather, a combination of different ERI is required to produce high technological innovation outcomes. Six configurations lead to higher innovation output in agribusiness and three configurations of non-high innovation output. It is impossible to blindly apply high intensity to all environmental regulation instruments without compromising the firm's technological innovation output. The agri-firms will not be able to achieve high innovation output if all environmental regulation instruments are implemented with a weak intensity.

2. Technological innovation in agribusiness cannot be solely facilitated by tightly controlled, command-based environmental regulation instruments. Intensive market-incentivized ERI carry out a dominant role in the high technological innovation performance of agribusinesses. Strong implicit environmental regulatory instruments play an important role in high innovation output, and their presence or absence is directly related to the ability of firms to achieve high innovation output. Voluntary ERI do not perform a significant role in promoting firms' technological innovation.

3. High environmental taxes and high environmental subsidies are potential substitutes that must cooperate. They are market-driven regulatory tools that function in opposition to one another. Businesses won't be able to achieve high innovation outcomes if neither is high-intensity.

### Policy recommendations

1. The government should pay attention to the synergy of various environmental regulation instruments. Local governments should fully consider the interaction between various ERI, and the actual situation of the region, balance the strength of various environmental regulatory instruments, avoid over-reliance on one type of ERI, and achieve the best incentive effect of environmental regulatory instruments on technological innovation in agricultural enterprises as a whole.

2. Market-incentivized ERI performs an obvious role in promoting technological innovation in agricultural enterprises. It is recommended to further improve the types and intensity of implementation of market-incentivized environmental regulations, such as carbon sink trading for agricultural products and agricultural green credit, and gradually transform the current environmental regulation system, mainly based on administrative orders and supplemented by market-incentivized, into one in which market-incentivized are the mainstay and administrative orders are supplemented, to give full play to the market's decisive role in agricultural environmental protection and technological innovation in agricultural production.

3. Build an all-around agricultural environmental governance network led by the government, with the primary body of enterprises and public participation. It is recommended that the government set up effective information transmission and problem feedback channels, such as the establishment of agricultural environmental protection classes, agricultural ecological and environmental problems reporting platforms and other measures to encourage the public, enterprises, industry organizations and other institutions to participate in the construction of the local agricultural environmental regulation system, strengthen the attention and supervision of all parties to the environmental protection behavior of agricultural enterprises, improve the intensity of implicit environmental regulation, to force agricultural enterprises to technological innovation and This will force agricultural enterprises to innovate and fulfill their environmental responsibilities.

4. Further improve the implementation of voluntary environmental regulation instruments. Currently, voluntary ERI is not effective in promoting technological innovation in agricultural enterprises. Consideration should be given to strengthening the evaluation of green technology innovation and other related content and adjusting the weighting of related indicators to improve the relevance and general impact of voluntary environmental certification and environmental labelling on enterprise technology innovation.

This paper studied the impact of environmental regulation on corporate innovation by limiting it to listed agricultural firms in China. Future research can replicate this study for other nations, locations, or sectors. In addition, while there are a variety of environmental regulation tools, future studies could draw on the same methodology as this paper and adopt other observables for further validation. Finally, researchers are urged to expand on this article to look into the best way to optimize a particular environmental regulatory instrument.

## Supporting information

**S1 File.**
(ZIP)

## Author Contributions

**Conceptualization:** Jinglin Xia, Liguo Zhang.

**Data curation:** Liguo Zhang.

**Formal analysis:** Jinglin Xia.

**Funding acquisition:** Liguo Zhang.

**Investigation:** Yuwei Song.

**Methodology:** Jinglin Xia.

**Resources:** Jinglin Xia, Yuwei Song.

**Software:** Jinglin Xia.

**Supervision:** Liguo Zhang.

**Validation:** Liguo Zhang, Yuwei Song.

**Writing – original draft:** Jinglin Xia.

**Writing – review & editing:** Liguo Zhang, Yuwei Song.

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
