## [Decision Letter · Decision Letter 0]

28 Sep 2023

PONE-D-23-26959The impact of environmental regulatory instruments on agribusiness technology innovation - A study of configuration effects based on fsQCAPLOS ONE

Dear Dr. Xia,

Thank you for submitting your manuscript to PLOS ONE. After careful consideration, we feel that it has merit but does not fully meet PLOS ONE’s publication criteria as it currently stands. Therefore, we invite you to submit a revised version of the manuscript that addresses the points raised during the review process.

We look forward to receiving your revised manuscript.

Kind regards,

Bing Xue, Ph.D.

Academic Editor

PLOS ONE

Journal Requirements:

    "We acknowledge the helpful comments from the anonymous reviewers and the financial support from the National Natural Science Foundation of China (Grant No. 72073054), Jiangxi Provincial Social Science "14th Five-Year Plan" (2022) Fund Project (Grant No. 22WT60), Jiangxi Provincial Social Science Foundation Key Project (Grant No. 22ZXQH07), and Jiangxi Provincial Specific Funds Project for Postgraduate Innovation (Grant No. YC2022-B135)."

   "We acknowledge the financial support from:

National Natural Science Foundation of China (Grant No. 72073054) for L. Zhang;

Jiangxi Provincial Social Science "14th Five-Year Plan" (2022) Fund Project (Grant No. 22WT60) for J. Xia, & L. Zhang;

Jiangxi Provincial Social Science Foundation Key Project (Grant No. 22ZXQH07) for L. Zhang;

Jiangxi Provincial Specific Funds Project for Postgraduate Innovation (Grant No. YC2022-B135) for J. Xia.

Reviewers' comments:

Reviewer's Responses to Questions

**Comments to the Author**

1. Is the manuscript technically sound, and do the data support the conclusions?

Reviewer #1: Yes

Reviewer #2: Yes

2. Has the statistical analysis been performed appropriately and rigorously? 

Reviewer #1: Yes

Reviewer #2: Yes

3. Have the authors made all data underlying the findings in their manuscript fully available?

Reviewer #1: Yes

Reviewer #2: Yes

4. Is the manuscript presented in an intelligible fashion and written in standard English?

Reviewer #1: Yes

Reviewer #2: Yes

5. Review Comments to the Author

Reviewer #1: Dear author,

After carefully examining your manuscript, we believe that this is a well-written paper containing interesting results that merit publication. For the benefit of the readers, however, a number of points need clarifying and certain statements require further justification. They are given as below:

1. Introduction: Agriculture-related businesses need to be more closely connected. It is essential to concentrate more on agricultural businesses than on other enterprises, taking into account the analysis of the reported outcomes.

2. Supporting Literature: The majority of the research discussed in this article relates to China, and it is advised that it be better applied in conjunction with pertinent international research.

3. Limitations and Further Research: It would be helpful if you discussed the limitations of your study more explicitly. The manuscript lacks directions on future research or the next steps following this study’s completion. Please consider providing some guidance in this area.

In conclusion, we think the subject matter covered in this paper is significant, the research techniques employed are innovative, and the article can be considered publishable after appropriate minor revisions. Look forward to reading your essay once it has been revised.

Best regards

Reviewer #2: Dear author,

This study applies fuzzy set qualitative comparative analysis, a novel methodology, to examine the causal links between environmental regulatory tools and technical innovation in agribusiness. The research was meticulously conducted, and the results are intriguing. Here are some suggested minor revisions:

1. Introduction: The application of the QCA method here is relatively novel. But the way the questions are raised should be smoother. After reviewing the current literature on environmental regulation and enterprise innovation, it is recommended to emphasize that previous literature did not consider the issue of configuration, in order to raise your question and innovation.

2. Methodology and Organization: It is unnecessary to give a thorough explanation of the QCA method's introduction as is done in textbooks. Please be kindly suggested to highlight the features of the fsQCA that were used and the justifications for its applicability.

3. Clarity and Accuracy: It should be mentioned that the manuscript contains a few grammatical and typographical errors. Term consistency is required for the entire text.

In conclusion, the topic discussed in this article is worth attention and the method of research is innovative. However, several minor issues must be addressed before publication. We appreciate your hard work and look forward to receiving your revised manuscript.

Sincerely,

Wang Tuzhan

6. PLOS authors have the option to publish the peer review history of their article (what does this mean?). If published, this will include your full peer review and any attached files.

Reviewer #1: No

Reviewer #2: No

---

## [Author Response · Author response to Decision Letter 0]

24 Oct 2023

Dear editors and reviewers,

We sincerely appreciate your valuable feedbacks, which we utilized to improve the quality of our manuscript. The reviewer comments are presented in italics below, followed by our response in regular font and changes to the manuscript in blue text. The key corrections in the paper, along with the responds to the reviewer’s comments are as flowing:

Reviewer #1

1. Introduction: Agriculture-related businesses need to be more closely connected. It is essential to concentrate more on agricultural businesses than on other enterprises, taking into account the analysis of the reported outcomes.

Thank you for the valuable advice. One of the reasons we chose to focus on agriculture in this paper is because there aren’t as many research on the connection between environmental regulation and technical innovation in the context of agribusiness. 

Nevertheless, please refer to line 41, 44, 77, and 127 in the INTRODUCTION part (in the marked manuscript) and REFERENCE 1, 2, 11, and 23 for the addition of new pertinent publications.

2. Supporting Literature: The majority of the research discussed in this article relates to China, and it is advised that it be better applied in conjunction with pertinent international research.

We sincerely appreciate the valuable comments. We have checked the literature carefully and added more references on the relevant research conducted in the countries and regions other than China in the revised manuscript.

Please refer to the revisions in the manuscript, mainly the line 72, 74, and 131 in the INTRODUCTION part and REFERENCE 5,7, 9, and 25.

3. Limitations and Further Research: It would be helpful if you discussed the limitations of your study more explicitly. The manuscript lacks directions on future research or the next steps following this study’s completion. Please consider providing some guidance in this area.

Thanks for pointing it out. We have added a paragraph indicating the limitations of this paper and the future direction of research.

Please kindly refer to line 662 to 667 in the updated manuscript.

 

Reviewer #2

1. Introduction: The application of the QCA method here is relatively novel. But the way the questions are raised should be smoother. After reviewing the current literature on environmental regulation and enterprise innovation, it is recommended to emphasize that previous literature did not consider the issue of configuration, in order to raise your question and innovation.

Thanks for the valuable suggestions. We have added summary about methodology of previous studies and raised the necessity of applying QCA method. 

Please kindly refer to line 138-153, and 162 in the INTRODUCTION part and REFERENCE 27 and 28.

2. Methodology and Organization: It is unnecessary to give a thorough explanation of the QCA method's introduction as is done in textbooks. Please be kindly suggested to highlight the features of the fsQCA that were used and the justifications for its applicability.

We have reorganized the RESEARCH METHODOLOGY part, emphasized more on the applicability of fsQCA.

Please kindly see the amended line 178-187 and 230-233.

3. Clarity and Accuracy: It should be mentioned that the manuscript contains a few grammatical and typographical errors. Term consistency is required for the entire text.

We tried our best to improve the manuscript and made some adjustments to the entire manuscript, including paraphrasing sentences to improve the content flow and fixing typos and grammatical errors. These changes will not influence the structure and substance of the paper. Please kindly see the change tracks in the documents as we couldn’t list all the amendments here. We earnestly appreciate the thoughtful effort of the editors and reviewers and anticipate that the correction will be well received.

Furthermore, we have polished the typesetting of the manuscript again to meet PLOS ONE's style guidelines, deleted the acknowledgement about financial support in the manuscript, and updated the funding statement in the cover letter.

Thank you very much for your attention and time. Looking forward to hearing from you.

Yours sincerely,

Jinglin Xia

---

## [Editor Report · Decision Letter 1]

7 Nov 2023

The impact of environmental regulatory instruments on agribusiness technology innovation - A study of configuration effects based on fsQCA

PONE-D-23-26959R1

Dear Dr. Xia,

We’re pleased to inform you that your manuscript has been judged scientifically suitable for publication and will be formally accepted for publication once it meets all outstanding technical requirements.

Kind regards,

Bing Xue, Ph.D.

Academic Editor

PLOS ONE
---

## [Editor Report · Acceptance letter]

8 Jan 2024

PONE-D-23-26959R1 

PLOS ONE

Dear Dr. Xia, 

I'm pleased to inform you that your manuscript has been deemed suitable for publication in PLOS ONE. Congratulations! Your manuscript is now being handed over to our production team.

Kind regards, 

on behalf of

Professor Bing Xue 

Academic Editor

PLOS ONE